

# The combined effects of light intensity, temperature, and water potential on wall deposition in regulating hypocotyl elongation of *Brassica rapa*

Hongfei Wang and  Qingmao Shang

Key Laboratory of Horticultural Crop Biology and Germplasm Innovation (Ministry of Agriculture), Institute of Vegetables and Flowers, Chinese Academy of Agricultural Sciences, Beijing, China

## ABSTRACT

Hypocotyl elongation is a critical sign of seed germination and seedling growth, and it is regulated by multi-environmental factors. Light, temperature, and water potential are the major environmental stimuli, and their regulatory mechanism on hypocotyl growth has been extensively studied at molecular level. However, the converged point in signaling process of light, temperature, and water potential on modulating hypocotyl elongation is still unclear. In the present study, we found cell wall was the co-target of the three environmental factors in regulating hypocotyl elongation by analyzing the extension kinetics of hypocotyl and the changes in hypocotyl cell wall of *Brassica rapa* under the combined effects of light intensity, temperature, and water potential. The three environmental factors regulated hypocotyl cell elongation both in isolation and in combination. Cell walls thickened, maintained, or thinned depending on growth conditions and developmental stages during hypocotyl elongation. Further analysis revealed that the imbalance in wall deposition and hypocotyl elongation led to dynamic changes in wall thickness. Low light repressed wall deposition by influencing the accumulation of cellulose, hemicellulose, and pectin; high temperature and high water potential had significant effects on pectin accumulation overall. It was concluded that wall deposition was tightly controlled during hypocotyl elongation, and low light, high temperature, and high water potential promoted hypocotyl elongation by repressing wall deposition, especially the deposition of pectin.

## INTRODUCTION

Light, temperature and water potential are the basic environmental conditions for plant survival and have important influences on plant growth and development, including seed germination, root initiation, hypocotyl elongation, leaf growth, flowering and so on *Hersch et al. (2014)*, *Lorenzo et al. (2016)*, *Patel & Franklin (2014)* and *Wilkinson & Davies (2010)*. These environmental factors are often correlated under natural conditions, and their combined effects on seedling growth are often analyzed using hypocotyl elongation, which has been an exemplar model system to study cell expansion (*Derbyshire et al., 2007*; *Refregier et al., 2004*). The synergistic regulation of two environmental factors on hypocotyl

Corresponding author
Qingmao Shang,
shangqingmao@caas.cn

elongation, such as "light intensity and temperature" and "light quality and temperature", has been analyzed (*Johansson et al., 2014*; *Kurepin et al., 2010*).

Hypocotyl is an appropriate system to study environmental control of plant growth in the early period. The cellular basis of hypocotyl elongation has been extensively studied, indicating that the contribution of cell elongation and division to hypocotyl growth is species- and growth condition-dependent (*Gendreau et al., 1997*; *Raz & Koornneef, 2001*; *Scheres et al., 1994*). For example, hypocotyl growth exclusively occurs through cell elongation with no contribution from cell division in *Arabidopsis thaliana* (*Boron & Vissenberg, 2014*; *Derbyshire, McCann & Roberts, 2007*). Cell elongation and division are both observed in elongating hypocotyl of *Helianthus annuus* (*Kutschera & Niklas, 2013*). Hypocotyl extension of *Cucumis sativus* occurs through cell elongation without division when grown in low light. However, cell elongation and division both contributed to hypocotyl growth under high light (*Lopez-Juez et al., 1995*).

The extensibility of primary wall plays a critical role in regulating hypocotyl cell elongation induced by environmental factors (*Cosgrove, 2016*; *Pereyra et al., 2010*; *Xiao, Somerville & Anderson, 2014*). The extensibility of the cell wall often reduces as cell wall polymers are deposited, which is necessary in elongating cells; otherwise, an increase in cell size would be achieved by stretching the existing wall (*Derbyshire et al., 2007*; *Refregier et al., 2004*). To protect the wall from excessive thinning and being ruptured by turgor pressure, cell expansion and wall biosynthesis are coordinately regulated (*Ivakov et al., 2017*; *Voxeur & Hofte, 2016*). The imbalance of cell elongation and wall biosynthesis contributes to dynamic changes in wall thickness (*Bischoff et al., 2011*; *Wu et al., 2005*; *Xiao et al., 2017*). As hypocotyl elongates, cell walls display phases of thickening, maintaining a constant thickness, or becoming thinner, depending on the cell type and developmental stage (*Derbyshire et al., 2007*; *Refregier et al., 2004*). The regulatory effect of single environmental factor on the deposition of cellulose, hemicellulose, and pectin has been extensively studied (*Derbyshire et al., 2007*; *Le Gall et al., 2015*; *Sasidharan, Voesenek & Pierik, 2011*). However, whether these components are coordinately regulated by multiple environmental factors should be further investigated.

*Brassica rapa* is an economically important vegetable that is popular worldwide. The hypocotyl of *B. rapa* is particularly sensitive to environmental conditions and can become over-elongated, resulting in the potential failure of transplants in the production process (*Devlin et al., 1997*; *Procko et al., 2014*). In this paper, the characteristics of hypocotyl growth, dynamic changes in wall thickness, and changes in wall compositions were measured to investigate the co-target of light intensity, temperature, and water potential in regulating hypocotyl elongation. Our hypothesis was that the cell wall was the co-target. Low light, high temperature, and high water potential changed wall properties by adjusting its composition and modulated hypocotyl cell growth.

## MATERIALS & METHODS

### Plant materials and treatments

Seeds of *B. rapa* (cv. CuiBai No. 3) were surface sterilized in 5% NaClO. The sterilized seeds were sown in vermiculite irrigated with 200 mL 1/2-strength Hoagland solution after

**Table 1  Growth conditions of *B. rapa* seedlings.**  Data in the table represent the means ± SE.

| Growth condition | Light intensity ($\mu$mol m$^{-2}$ s$^{-1}$) | Temperature (°C) | Water potential (MPa) |
|---|---|---|---|
| $L_h T_l W_l$ | 250 ± 10 | 21 ± 1 | −0.15 ± 0.01 |
| $L_h T_l W_h$ | 250 ± 10 | 21 ± 1 | −0.05 ± 0.01 |
| $L_h T_h W_l$ | 250 ± 10 | 29 ± 1 | −0.15 ± 0.01 |
| $L_h T_h W_h$ | 250 ± 10 | 29 ± 1 | −0.05 ± 0.01 |
| $L_l T_l W_l$ | 50 ± 10 | 21 ± 1 | −0.15 ± 0.01 |
| $L_l T_l W_h$ | 50 ± 10 | 21 ± 1 | −0.05 ± 0.01 |
| $L_l T_h W_l$ | 50 ± 10 | 29 ± 1 | −0.15 ± 0.01 |
| $L_l T_h W_h$ | 50 ± 10 | 29 ± 1 | −0.05 ± 0.01 |

**Notes.**

Abbreviations: L, light intensity; T, temperature; W, water potential; h, high; l, low.

germination, and then cultured at 25 °C in the dark. At the onset of seedling emergence, the seedlings were irrigated with another 200 mL 1/2-strength Hoagland solution or a solution with 8% polyethylene glycol 6000 (PEG-6000; w/v; Sinopharm, Beijing, China), and then cultured in controlled chambers equipped with fluorescent lights (Philips, 28 W, Amsterdam, Netherlands). The light intensity was set to 50 and 250 $\mu$mol m$^{-2}$ s$^{-1}$ (16 h light/8 h dark, Table 1), and the R:FR ratios in the different treatments were all 4:1. The spectral outputs were shown in Fig. S1. The temperature was set to 21 °C and 29 °C, which were kept constant day and night. The water potential of irrigation solutions without or with PEG-6000 was measured by a Psypro water potential system (Wescor, Logan, KY, USA).

## Measurements of hypocotyl length, elongation rate, volume and cell length

The hypocotyl length of *B. rapa* seedlings was measured in 1 d intervals using a millimeter scale (accuracy ± 0.5 mm), and 30 plants were scored in three independent replicates of each treatment. Adjacent measurements were used to calculate the elongation rate. Hypocotyls treated for 8 d were cut into small segments at the midpoint and fixed in 37% formaldehyde: acetic acid: ethanol: water (5:5:63:27) for 2 d at 4 °C. Then, the segments were gradually dehydrated in a series of alcohol solutions, incubated in ethylbenzene/paraffin at 58 °C, and embedded in paraffin. The samples were sliced into 9 $\mu$m sections and stained using Fast Green dye. The lengths of cells in the epidermis and cortex were measured and photographed using a BX53 light microscope (Olympus, Tokyo, Japan). In total, 15 sections that included 75 cells in three independent replicates were used for cell length determination. Cell number of epidermis and cortex was calculated from the hypocotyl length divided by the cell length.

## Measurement of cell wall thickness

After the seedlings were treated for 0, 2, 5, or 8 d, they were cut into 2 mm segments at the midpoint and fixed in 3% glutaraldehyde buffer (v/v; pH 7.3) in the dark. The segments were rinsed with phosphatic buffer solution (0.1 M, pH 7.2) and post-fixed in 1% OsO$_4$, which was followed by washes with phosphatic buffer solution. Then, the samples were

gradually dehydrated in a series of alcohol solutions, incubated in acetone/resin at 35 °C and embedded in resin at 40 °C. The samples were sliced into ultrathin transverse sections of ∼90 nm using a glass knife on a Reichert ultramicrotome (Leica, Milton Keynes, UK). The slices were collected on 200-mesh copper grids and stained with 2% (w/v) uranyl acetate for 15 min and 1% (w/v) lead citrate for 10 min. Then, the slices were washed with water and air-dried in petri dishes. Cell walls were observed using a Jeol JEM-1230 transmission electron microscope (JEOL, Tokyo, Japan) and photographed. Envisioning the hypocotyls as cylinders, the volume = $\pi$ * (radius of hypocotyl at midpoint)$^2$ * hypocotyl length. The wall volume was calculated from sections at the midpoint of the hypocotyl, and the transverse wall was not included (Fig. S2A), as previously described (*Derbyshire et al., 2007*). The wall volume of the outer epidermal wall (OE) was calculated by multiplying the perimeter of the hypocotyl at the midpoint by the OE thickness and by the hypocotyl length. The volume of the inner epidermal wall (IE) was calculated by multiplying the perimeter of the IE by the number of epidermis, by the thickness of the IE, and by the hypocotyl length. The wall volume of the cortical wall (CO) adjacent to the OE was considered the CO volume and calculated by multiplying the perimeter of the CO by its number, by the thickness of the CO, and by the hypocotyl length (Fig. S2B). The perimeter of the IE = 2 $\pi$ * radius of epidermis cell * the cell number − the perimeter of the hypocotyl at the midpoint. The number of epidermal cell = 2 $\pi$ * (radius of hypocotyl − the radius of epidermal cell)/diameter of the epidermal cells. The number of cortical cell = 2 $\pi$ * (the radius of hypocotyl − the diameter of epidermal cell − the radius of cortical cell adjacent to the epidermal cell)/diameter of the cortical cells. The data used to calculate the index were from three independent replicates that included 6 hypocotyls and 60 cells. The fold change in hypocotyl volume induced by low light = $(L_lT_lW_l/L_hT_lW_l + L_lT_lW_h/L_hT_lW_h + L_lT_hW_l/L_hT_hW_l + L_lT_hW_h/L_hT_hW_h)/4$; the fold change in hypocotyl volume induced by high temperature = $(L_hT_hW_l/L_hT_lW_l + L_hT_hW_h/L_hT_lW_h + L_lT_hW_l/L_lT_lW_l + L_lT_hW_h/L_lT_lW_h)/4$; the fold change in hypocotyl volume induced by high water potential = $(L_hT_lW_h/L_hT_lW_l + L_hT_hW_h/L_hT_hW_l + L_lT_lW_h/L_lT_lW_l + L_lT_hW_h/L_lT_hW_l)/4$.

## Determination of hypocotyl cell wall mass and content in components

The cell wall mass of hypocotyls was determined according to the previous method (*Zhong & Lauchli, 1993*). Approximately 40 hypocotyl segments treated for 8 d were harvested, weighed, and frozen in liquid nitrogen. The frozen samples were homogenized to powder and washed into centrifuge tubes using 1 mL ice-cold 75% ethanol. The tubes were kept in ice for 20 min without disturbing, followed by centrifugation for 10 min at 10,000 × g. The pellets were sequentially washed using ice-cold acetone, a mixture of methanol and chloroform (1:1, v/v), and methanol. The pellets were considered as cell wall preparations and weighed after lyophilization.

The cell wall preparations were fractionated into four fractions, including pectin, hemicellulose 1 (HC1), hemicellulose 2 (HC2), and cellulose (*Iraki et al., 1989*; *Labavitch & Ray, 1974*). The freeze-dried pellets were suspended in 2 mL 0.5% ammonium oxalate

buffer containing 0.1% $NaBH_4$ (pH 4.0) in a boiling water bath for pectin extraction. Then, the pellets were subjected to another extraction with 5 mL 4% KOH containing 0.1% $NaBH_4$ at room temperature, and the extraction solutions were HC1 fractions. The protocol for the HC2 extraction was similar to that for HC1 extraction, but 24% KOH was used. Before determination, HC1 and HC2 solutions were neutralized with glacial acetic acid. The residuals in the centrifuge tubes were regarded as cellulose and freeze-dried before weighing. The pectin content was determined by carbazole colorimetry as previously described (*Stark, 1950*). Hemicellulose content was determined by anthrone assay (*Updegraff, 1969*). The decrease in wall component content induced by low light = $(1 - \text{average} (L_lT_lW_l/L_hT_lW_l + L_lT_lW_h/L_hT_lW_h + L_lT_hW_l/L_hT_hW_l + L_lT_hW_h/L_hT_hW_h)) * 100\%$; the decrease in wall component content induced by high temperature = $(1 - \text{average} (L_hT_hW_l/L_hT_lW_l + L_hT_hW_h/L_hT_lW_h + L_lT_hW_l/L_lT_lW_l + L_lT_hW_h/L_lT_lW_h)) * 100\%$; the decrease in wall component content induced by high water potential = $(1 - \text{average} (L_hT_lW_h/L_hT_lW_l + L_hT_hW_h/L_hT_hW_l + L_lT_lW_h/L_lT_lW_l + L_lT_hW_h/L_lT_hW_l)) * 100\%$.

### Gene expression analysis by quantitative RT-PCR

Total RNA was extracted from hypocotyls using TRIzol reagent (Invitrogen, Gaithersburg, MD, USA) according to the manufacturer's instruction. The first-strand cDNA was synthesized using a reverse transcription system (Promega, Madison, WI, USA), and real-time PCR was carried out with a SYBR Green Supermix (Transgene, Beijing, China) on a LightCycler® 96 real-time PCR system (Roche, Basel, Switzerland). The reactions were performed with three replicates using *GAPDH* (*Bra016729*) as the reference gene (*Procko et al., 2014*; *Qi et al., 2010*), and the relative expression levels of the target genes were calculated using the $2^{-\Delta\Delta Ct}$ method (*Schmittgen & Livak, 2008*). The primers used to quantify the gene expression levels were listed in Table S1.

### Statistical analysis

The values presented in pictures and tables represent the mean (three replicates) $\pm$ standard error (SE). Significance was analyzed using SAS 9.2.0 software with Duncan's multiple range test ($P < 0.05$).

## RESULTS

### Hypocotyl elongation kinetics

Hypocotyl elongation kinetics of *B. rapa* seedlings were established in eight treatments that were used to detect the combined effects of light intensity, temperature, and water potential on hypocotyl growth. Hypocotyl length was measured over a period of 8 d, and the final length showed a high degree of variability in the eight treatments: high light × low temperature × low water potential ($L_hT_lW_l$), high light × low temperature × high water potential ($L_hT_lW_h$), high light × high temperature × low water potential ($L_hT_hW_l$), high light × high temperature × high water potential ($L_hT_hW_h$), low light × low temperature × low water potential ($L_lT_lW_l$), low light × low temperature × high water potential ($L_lT_lW_h$), low light × high temperature × low water potential ($L_lT_hW_l$), and
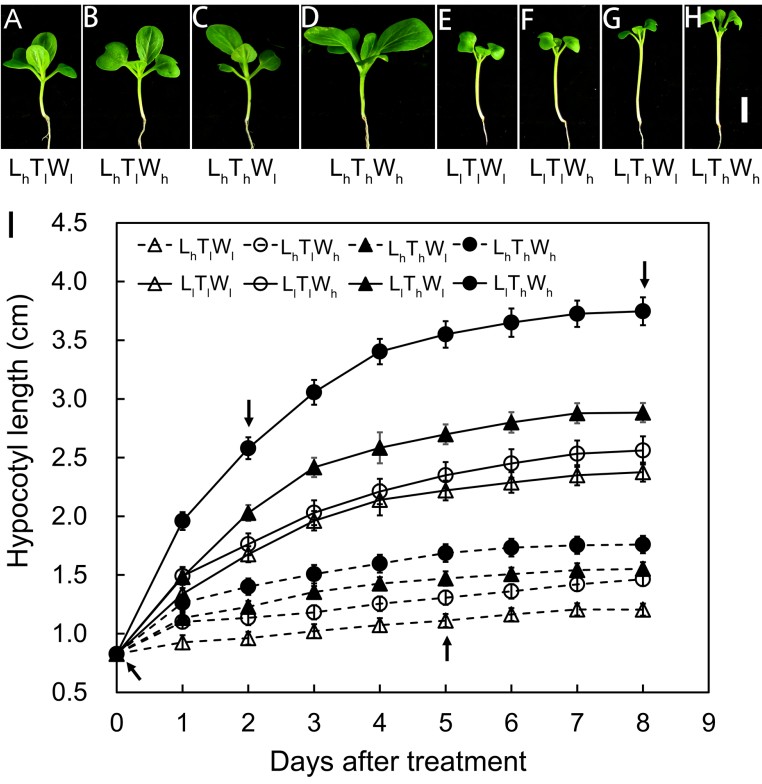

**Figure 1** **Hypocotyl elongation dynamics in *B. rapa* responding to light intensity, temperature and water potential.** (A–H) Photographs of *B. rapa* seedling under different growth conditions at 8 d. Scale bars = 1 cm. (I) Dynamic curve of hypocotyl elongation. Values in the pictures are means ± SE ($n = 30$). Arrows indicate times (0, 2, 5, and 8 d) at which hypocotyls are used to analyze wall thickness in further. Abbreviations: L represents light intensity; T represents temperature; W represents water potential; h represents high; l represents low.

low light × high temperature × high water potential ($L_lT_hW_h$) (Figs. 1A–1H). According to binary comparisons ($L_lT_lW_l$ vs $L_hT_lW_l$, $L_lT_lW_h$ vs $L_hT_lW_h$, $L_lT_hW_l$ vs $L_hT_hW_l$, and $L_lT_hW_h$ vs $L_hT_hW_h$), the increase in hypocotyl length induced by low light was 1.17, 1.10, 1.33, and 1.99 cm, respectively. The effect of light intensity on hypocotyl growth was affected by temperature and water potential. The increase in hypocotyl length induced by high temperature ($L_hT_hW_l$ vs $L_hT_lW_l$, $L_hT_hW_h$ vs $L_hT_lW_h$, $L_lT_hW_l$ vs $L_lT_lW_l$, and $L_lT_hW_h$ vs $L_lT_lW_h$) were 0.35, 0.30, 0.51, and 1.19 cm, respectively. The promotion of high temperature on hypocotyl growth was highly dependent on light intensity and water potential. The hypocotyl length increased by 0.26, 0.21, 0.19, and 0.86 cm when grown under the high water potential conditions ($L_hT_lW_h$ vs $L_hT_lW_l$, $L_hT_hW_h$ vs $L_hT_hW_l$, $L_lT_lW_h$ vs $L_lT_lW_l$, and $L_lT_hW_h$ vs $L_lT_hW_l$). The influence of high water potential was tightly dependent on light intensity and temperature. The results above indicated that light intensity, temperature, and water potential regulated hypocotyl elongation independently and in coordination.

The hypocotyls of *B. rapa* seedlings grown under the eight conditions above were analyzed in further at equivalent developmental stages, including stage I at the onset of seedling emergence, stage II at 50% of the maximal increase in hypocotyl length, stage III at 90% of the maximal increase in hypocotyl length, and stage IV at the final hypocotyl length, which were estimated from the elongation curves and indicated by the arrows in Fig. 1B. These stages were set at 0, 2, 5, and 8 d after treatments, and the general morphology of the seedlings at the four stages above were shown in Figs. 1A–1H and Fig. S3.

## Cell elongation primarily contributed to hypocotyl extension

Next, the changes in cell length and the number of cell files in the longitudinal direction accompanying hypocotyl elongation were examined. The structure of the *B. rapa* hypocotyl was relatively simple, mainly consisting of epidermis, cortex, endodermis, and pericycle (Fig. S4). Epidermis is the outermost cell that determines the elongation rates of organs (*Kutschera & Niklas, 2007*). Cortical cells are relatively abundant in the *B. rapa* hypocotyl. Therefore, the epidermis and cortex were selected for further analysis. The length of epidermal and cortical cells was measured using paraffin sections at the midpoint of the hypocotyl (Fig. 2), and cell numbers in the longitudinal direction were calculated according to cell and hypocotyl length (Table 2). The midpoint was selected to measure the cell length because hypocotyl elongation after seedling germination was mainly caused by the middle and apical segments, and the cell length at the midpoint was similar to the average length of hypocotyl cells (*Gendreau et al., 1997*; *Procko et al., 2014*; *Paque et al., 2014*). In addition, light intensity, temperature, and water potential regulated hypocotyl elongation mainly through affecting the growth of middle and apical segments (Fig. S5). Under the combined effects of light intensity, temperature, and water potential, the difference in cell lengths of epidermal and cortical cells reached a significant level among the eight treatments ($P < 0.05$; Duncan's multiple range test). According to binary comparisons (all the binary comparison below were done in the same order as the corresponding comparisons in the part of Hypocotyl elongation kinetics), the lengths of epidermal cells increased by 31.91, 76.30, 21.58, and 27.31 µm under low light, and the corresponding increases in cortical cell length were 64.65, 79.73, 49.51, 55.91 µm. High temperature promoted epidermal cell elongating by 10.27, 12.60, 20.60, and 61.59 µm, and cortical cell elongating by 14.75, 15.83, 29.89, and 39.64 µm. Under the influence of high water potential, epidermal cells elongated by 6.57, 8.90, 12.30, and 53.27 µm, and cortical cells elongated by 15.24, 16.31, 21.63, and 31.40 µm. Further analysis indicated that the influences of light intensity, temperature, and water potential on hypocotyl growth were all significant ($P < 0.05$; Duncan's multiple range test). In addition, the number of the epidermal and cortical cells on the longitudinal axis was calculated. About 50, 20, and 20 layers were added in epidermis responding to low light, high temperature, and high water potential. The corresponding variation in the number of cortical cell layers was approximately 20, 20, and 10. The influence of light intensity on epidermal cell layers reached a significant level, but not on the number of cortical cells ($P < 0.05$; Duncan's multiple range test). Temperature and water potential had no significant effects on the number of epidermal and cortical cell. To elucidate the contribution of cell division to hypocotyl elongation induced by low light,

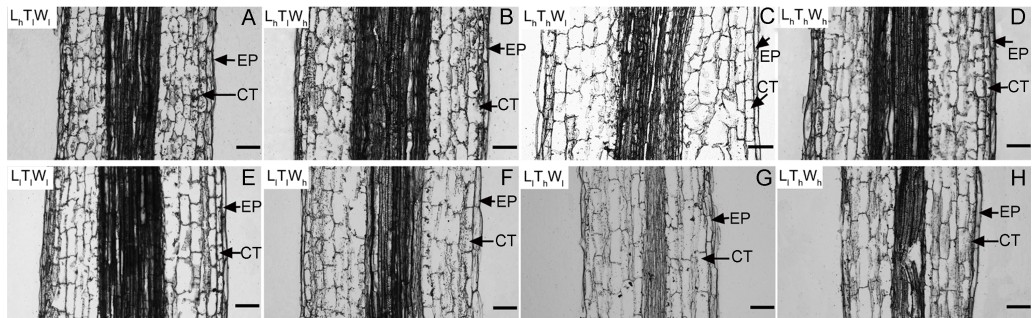

**Figure 2** **Pictures of paraffin section in hypocotyl of *B. rapa* treated for 8 d.** (A–H) Paraffin sections of *B. rapa* seedling grown under $L_hT_lW_l$, $L_hT_lW_h$, $L_hT_hW_l$, $L_hT_hW_h$, $L_lT_lW_l$, $L_lT_lW_h$, $L_lT_hW_l$, $L_lT_hW_h$. Scale bars = 100 μm. Abbreviations: L represents light intensity; T represents temperature; W represents water potential; h represents high; l represents low; EP represents epidermis; CT represents cortex.

**Table 2** **The combined effects of light intensity, temperature, and water potential on cell length and cell number in *B. rapa* hypocotyl.** Values in the table are means ± SE ($n = 15$). Different letters (a, b, c, d, e, and f) are used to indicate significance among treatments ($P < 0.05$; Duncan's multiple range test).

| Treatment | Epidermis | | Cortex | |
|---|---|---|---|---|
| | Cell length (μm) | Cell number | Cell length (μm) | Cell number |
| $L_hT_lW_l$ | 86.94 ± 2.32 f | 127.88 ± 1.32 d | 79.48 ± 8.67 f | 139.88 ± 1.23 de |
| $L_hT_lW_h$ | 93.51 ± 3.48 e | 139.84 ± 4.37 e | 94.72 ± 5.15 e | 138.58 ± 6.90 e |
| $L_hT_hW_l$ | 97.20 ± 5.53 e | 151.25 ± 8.15 c | 94.23 ± 4.54 e | 156.04 ± 7.08 bc |
| $L_hT_hW_h$ | 106.10 ± 4.16 d | 158.97 ± 5.18 c | 110.55 ± 8.41 d | 153.93 ± 10.69 cd |
| $L_lT_lW_l$ | 108.51 ± 7.39 d | 195.49 ± 2.45 b | 129.00 ± 3.88 c | 164.38 ± 3.27 bc |
| $L_lT_lW_h$ | 120.81 ± 4.72 c | 194.79 ± 3.69 b | 150.63 ± 10.20 b | 156.07 ± 8.66 bc |
| $L_lT_hW_l$ | 129.12 ± 3.17 b | 209.16 ± 3.63 a | 158.88 ± 6.02 b | 169.95 ± 4.58 ab |
| $L_lT_hW_h$ | 182.40 ± 11.47 a | 189.14 ± 2.95 b | 190.28 ± 6.22 a | 181.48 ± 4.22 a |

**Notes.**
Abbreviations: L, light intensity; T, temperature; W, water potential; h, high; l, low.

high temperature, and high water potential in further, the relative expression level of the marker genes: *CDCA3;2*, *CDCB1;1* and *CDKA;1*, was investigated (*Dewitte & Murray, 2003*; *Joubès et al., 2000*). And only light intensity have significant influences on the express level of genes involved in cell division ($P < 0.05$; Duncan's multiple range test) (Fig. S6). The results above implied that cell elongation primarily contributed to hypocotyl elongation in *B. rapa* induced by high temperature and high water potential. Cell division and elongation both contributed to hypocotyl elongation under the influence of low light.

## Dynamic changes in cell wall thickness

TEM was used to observe the cell wall of the epidermis and cortex at the four developmental stages indicated in Fig. 1. Typical micrographs of the OE, IE, CO in the hypocotyl at the final length (stage IV) were shown in Fig. 3.

At the onset of seedling emergence (stage I), the OE, IE, and CO were approximately 1.19, 0.50, and 0.36 μm, respectively (Fig. 4), and the thickness of the OE, IE, and CO thickened

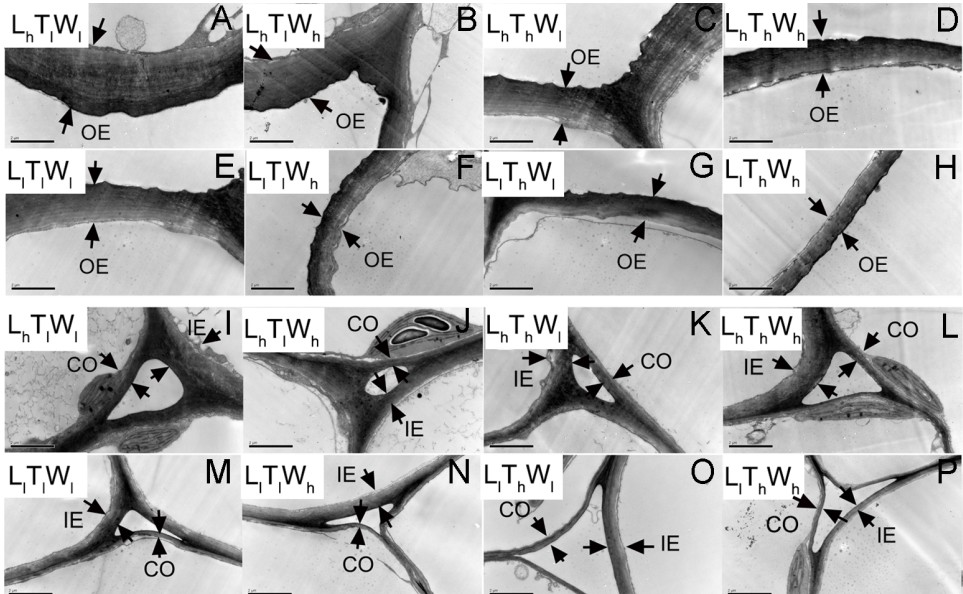

**Figure 3** **Transmission electron photographs of hypocotyl at the mid-point in *B. rapa* hypocotyl grown for 8 d.** (A–H) The micrographs of OE at the mid-point in *B. rapa* hypocotyl. (I–P) The micrographs of the cell corner at the mid-point in *B. rapa* hypocotyl. Arrows in the picture indicate the walls used for thickness measurement. Abbreviations: L, light intensity; T, temperature; W, water potential; h, high; l, low; OE, outer epidermal wall; IE, inner epidermal wall; CO, cortical wall. Scale bar = 2 μm.

by 24%, 14% and 5% on average, following seedling emergence (stage II). However, the OE thickened by 13%, and the IE and CO thinned by 5% and 25% at stage III, respectively. At stage IV, the OE and IE thickened by 43% and 16% on average, respectively. But the CO thinned by 4%, compared with the thickness at stage I.

According to binary comparisons, light intensity had significant effects on wall thickness at all three stages (Fig. 4, Figs. S7–S9 and Table S2) ($P < 0.05$; Duncan's multiple range test). The OE thinned by 21%, 38%, and 19% on average under the effect of low light at stage II, stage III, and stage IV, respectively. Similar responses of the IE and CO were observed: the IE thinned by 26%, 32%, and 35%, and the CO thinned by 25%, 29%, and 35% at the three stages. Compared with the hypocotyls grown at low temperature, those grown at high temperature acquired thinner cell walls. The thickness of the OE decreased by 12%, 21%, and 16% at stage II, stage III, and stage IV, respectively. And IE thinned by 18%, 18%, and 9% at stage II, stage III, and stage IV, respectively; the CO thinned by 15%, 22%, and 14% at stage II, stage III, and stage IV, respectively. Water potential also had a significant effect on the dynamic changes in wall thickness (Table S2; $P < 0.05$; Duncan's multiple range test), and the OE thinned by 11%, 11%, and 14% at stage II, stage III, and stage IV under the influence of high water potential, respectively. The thinning of inner walls was more dramatic, with a 17%, 20%, and 18% decrease in IE and a 14%, 17%, and 18% decrease in CO.

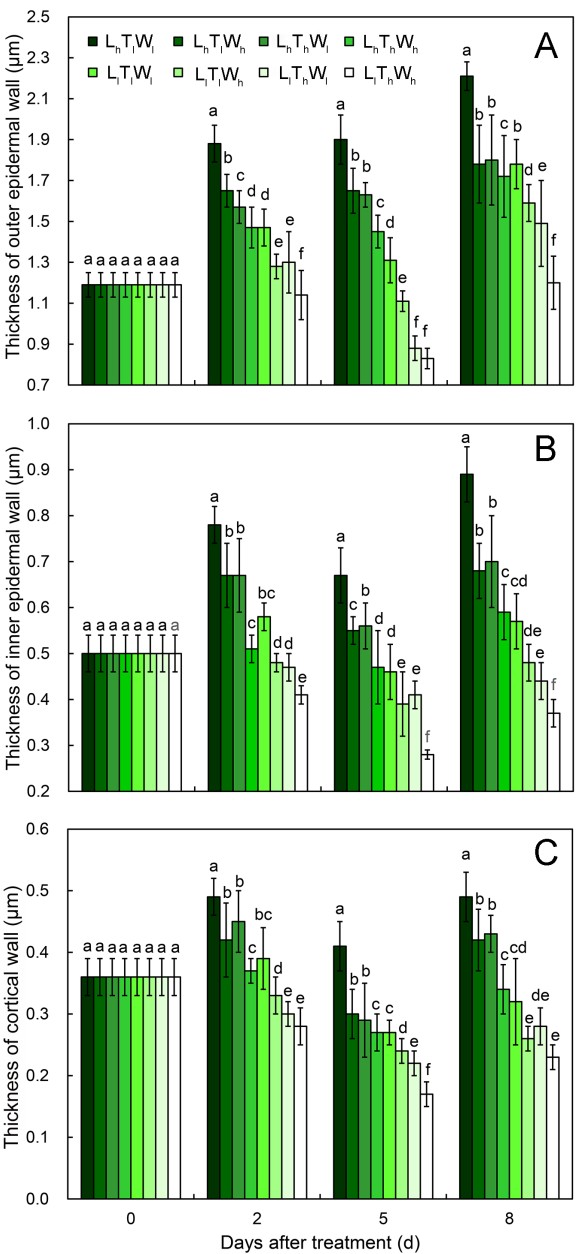

**Figure 4 Cell wall thickness of hypocotyls under the combined effects of light intensity, temperature, and water potential at different developmental stages.** (A) Dynamic change in thickness of outer epidermal wall. (B) Dynamic change in thickness of inner epidermal wall. (C) Thickness of cortical wall. Values in the figure are means $\pm$ SE ($n = 15$). Different letters (a, b, c, d, e, and f) are used to indicate significance among treatments ($P < 0.05$; Duncan's multiple range test).

As the hypocotyl elongated, the thickness of the OE changed with the trend of thickening-maintaining-thickening under the conditions of $L_hT_lW_l$, $L_hT_lW_h$, $L_hT_hW_l$, and $L_hT_hW_h$, and it changed with the trend of thickening-thinning-thickening under the conditions of $L_lT_lW_l$, $L_lT_lW_h$, $L_lT_hW_l$, and $L_lT_hW_h$. The thickness of the IE and CO changed by

thickening-thinning-thickening under $L_hT_lW_l$, $L_hT_lW_h$, $L_hT_hW_l$, $L_hT_hW_h$, and $L_lT_lW_l$ and changed by thinning-thinning-thickening under $L_lT_lW_h$, $L_lT_hW_l$, and $L_lT_hW_h$. Low light repressed OE, IE, and CO thickening at stage II, stage III, and stage IV. High temperature and high water potential had similar effects on wall thickness, but their effects were weaker than those of light intensity.

## Cell wall was the co-target of the three environmental factors in regulating hypocotyl elongation

The amount of wall deposition had important impacts on wall thickness, which was negatively related to wall extension (*Derbyshire et al., 2007*). We measured the wall mass and component contents in hypocotyls under the coordinated regulation of light intensity, temperature, and water potential (Table 3). The variation in wall mass was consistent with that of wall volume, which was calculated based on the wall thickness in Fig. 4, indicating that TEM could be used to measure wall thickness.

To further analyze the effects of light intensity, temperature, and water potential on wall deposition, the content of wall components was measured. The analysis of variance (ANOVA) about cellulose, hemicellulose, and pectin indicated that pectin was the co-target of environmental factors in regulating hypocotyl cell elongation (Table S3; $P < 0.05$; Duncan's multiple range test). Low light repressed cellulose, hemi-cellulose, and pectin deposition significantly (Table S3; $P < 0.05$; Duncan's multiple range test), and their contents decreased by 43%, 41%, and 43% on average, respectively. High temperature had effects on deposition of cellulose, hemicellulose, and pectin, inducing their contents decreased by 13%, 9%, and 15%, respectively. Further analysis indicated that the effect on the deposition of pectin reached significant level on the whole (Table S3; $P < 0.05$; Duncan's multiple range test). And the effect on hemicellulose reached significant level only under specific environmental factors ($L_lT_lW_l$ vs $L_lT_hW_l$, $L_lT_lW_h$ vs $L_lT_hW_h$; Table 3; $P < 0.05$; Duncan's multiple range test). High water potential inhibited wall deposition with decreases of 11%, 13%, and 15% in cellulose, hemicellulose, and pectin on average. Its effect on the deposition of cellulose reached significant level under specific conditions ($L_lT_lW_l$ vs $L_lT_lW_h$, $L_lT_hW_l$ vs $L_lT_hW_h$) (Table 3; $P < 0.05$; Duncan's multiple range test). Although its effect on pectin accumulation was significant on the overall level, it did not reach a significant level under specific conditions, such as $L_lT_lW_l$ vs $L_lT_lW_h$ (Table 3; $P < 0.05$; Duncan's multiple range test).

Hypocotyl cell elongation induced by low light, high temperature, and high water potential was mainly affected by wall extensibility, which was under the control of wall deposition and wall-modifying proteins, such as expansins (EXPA) and xyloglucan endotransglucosylase/hydrolase (XET/XTH). To investigate the change in wall extensibility as hypocotyl cells elongated, the expression levels of genes involved in cell wall biosynthesis and modification were investigated (Figs. S10–S11). Light intensity had a significant influence on the biosynthesis of wall components, and some genes involved in cellulose, hemicellulose, and pectin biosynthesis were upregulated by low light, such as *CesA6*, *CSLC4*, and *XXT5* (Table S4; $P < 0.05$; Duncan's multiple range test). Comparing with $L_hT_lW_h$, $L_hT_hW_l$, and $L_hT_hW_h$, the expression level of gene involved in pectin biosynthesis, *GAUT7*,

Wang and Shang (2020), *PeerJ*, DOI 10.7717/peerj.9106

**Table 3  Cell wall mass and component contents under the influence of light intensity, temperature and water potential.** Data in the table represent the means $\pm$ SE. The letters (a, b, c, d, e, and f) indicate significance among treatments ($P < 0.05$; Duncan's multiple range test).

| Content (mg g FW$^{-1}$) | $L_h T_l W_l$ | $L_h T_l W_h$ | $L_h T_h W_l$ | $L_h T_h W_h$ | $L_l T_l W_l$ | $L_l T_l W_h$ | $L_l T_h W_l$ | $L_l T_h W_h$ |
|---|---|---|---|---|---|---|---|---|
| Wall mass | 47.54 $\pm$ 2.03 a | 43.82 $\pm$ 1.10 b | 43.23 $\pm$ 1.20 b | 36.78 $\pm$ 1.45 c | 27.09 $\pm$ 0.90 d | 23.08 $\pm$ 0.47 e | 22.42 $\pm$ 1.02 e | 19.69 $\pm$ 0.75 f |
| Cellulose | 11.12 $\pm$ 0.26 ab | 11.93 $\pm$ 0.83 a | 10.65 $\pm$ 0.93 b | 10.56 $\pm$ 0.79 b | 8.29 $\pm$ 0.60 c | 5.69 $\pm$ 0.34 e | 6.23 $\pm$ 0.40 d | 4.94 $\pm$ 0.42 f |
| Hemicellulose | 1.34 $\pm$ 0.27 a | 1.26 $\pm$ 0.35 b | 1.33 $\pm$ 0.11 a | 1.21 $\pm$ 0.15 b | 0.89 $\pm$ 0.22 c | 0.75 $\pm$ 0.17 d | 0.77 $\pm$ 0.17 d | 0.62 $\pm$ 0.08 e |
| Pectin | 4.10 $\pm$ 0.45 a | 4.03 $\pm$ 0.26 a | 3.81 $\pm$ 0.34 b | 3.01 $\pm$ 0.17 c | 2.52 $\pm$ 0.23 d | 2.07 $\pm$ 0.34 e | 2.22 $\pm$ 0.35 e | 1.77 $\pm$ 0.38 f |

**Notes.**

Abbreviations: L, light intensity; T, temperature; W, water potential; h, high; l, low.
was higher in the condition of $L_lT_lW_h$, $L_lT_hW_l$, and $L_lT_hW_h$. Temperature had a significant effect on the expression level of genes involved in the biosynthesis of hemicellulose and pectin (Table S4; $P < 0.05$; Duncan's multiple range test). Under the influence of high temperature, the expression level of *XXT1*, *XXT5*, *MUR3*, *GAUT7*, and *RGX2* improved overall, and the expression level of *CSLC6* reduced. In addition, the expression level of the pectin biosynthesis gene, *GAUT1*, improved significantly under $L_hT_lW_l$ and $L_hT_lW_h$ compared with that under $L_hT_hW_l$, and $L_hT_hW_h$. Water potential had an effect on pectin synthesis, and the expression levels of *GAUT7* and *RGXT2* were upregulated by high water potential. The expression level of *GAUT7* and *RGXT2* was higher under $L_hT_lW_h$, $L_hT_hW_h$, $L_lT_lW_h$, and $L_lT_hW_h$ than that under $L_hT_lW_l$, $L_hT_hW_l$, $L_lT_lW_l$, and $L_lT_hW_l$, respectively. In summary, the biosynthesis of pectin was co-regulated by light intensity, temperature, and water potential. According to *Paque et al. (2014)*, five *XTHs* and one *EXPA* genes were selected to investigate their expression level under the combined effects of light intensity, temperature, and water potential (Fig. S11). Light intensity had a significant effect on the six genes (Table S4), and the expression of *XTH17*, *XTH22*, *XTH31*, *XTH33*, and *EXPA20* was induced by low light. High temperature had a significant effect on the expression of *XTH17*, *XTH18*, *XTH22*, and *EXPA20* (Table S4). While, only the expression level of *EXPA20* was significantly improved by high temperature. Water potential also had a significant effect on the target genes, and only the expression of *XTH17*, *XTH18*, and *XTH33* was significantly induced by high water potential in different treatments ($L_hT_lW_h$ vs $L_hT_lW_l$, $L_hT_hW_h$ vs $L_hT_hW_l$, $L_lT_lW_h$ vs $L_lT_lW_l$, $L_lT_hW_h$ vs $L_lT_hW_l$). The results indicated that the three environmental factors regulated hypocotyl elongation by changing the expression level of different genes involved in affecting wall extensity.

## Wall volume and hypocotyl volume are coordinately regulated

During hypocotyl elongation, the increases in the volume of the cell wall and hypocotyl were not always coordinated, which contributed to dynamic changes in wall thickness. Based on hypocotyl length and wall thickness, hypocotyl and wall volume were calculated (Table 4). The volume of the hypocotyl incrementally increased as it elongated, but the wall volume did not. At the final hypocotyl length, the hypocotyl volume increased by 1.92-, 1.29-, and 1.60-fold in response to low light, high temperature, and high water potential, respectively. The increases in the volume of the OE were 1.53-, 1.08-, and 1.18-fold, correspondingly; the increases in the volume of the IE were 1.68-, 1.05-, and 1.39-fold, correspondingly; and the increases in the volume of the CO were 1.34-, 1.06-, and 1.19-fold, correspondingly. The imbalance in the increases in the hypocotyl and cell wall volumes contributed to the dynamic changes in wall thickness under the combined effects of the three environmental factors. The analysis of wall volume further revealed that approximately 45% of the total wall volume was primarily present in the OE and approximately 30% in the IE. The other 25% presented in the CO. The results above were consistent with the epidermis-controlled organ elongation in a previous study (*Kutschera & Niklas, 2007*).

**Table 4** **Relationship analysis about the increase of cell wall volume and hypocotyl elongation.** Data in the table represent the means ± SE. Lowercase letters represent significance among treatments ($P < 0.05$; Duncan's multiple range test).

| Condition | Volume (mm³) | | | | | Cell number | |
|---|---|---|---|---|---|---|---|
| | HY | OE | IE | CO | TCW | EP | CO |
| $L_h T_l W_l$ | 4.799 f | 0.060 g | 0.030 h | 0.038 c | 0.128 g | 225.41 b | 227.49 a |
| $L_h T_l W_h$ | 6.237 e | 0.060 g | 0.039 f | 0.035 d | 0.134 f | 198.98 c | 166.78 c |
| $L_h T_h W_l$ | 6.540 e | 0.064 f | 0.035 g | 0.039 c | 0.138 f | 200.62 c | 197.53 b |
| $L_h T_h W_h$ | 8.608 d | 0.075 e | 0.040 e | 0.037 cd | 0.152 e | 176.32 d | 160.26 c |
| $L_l T_l W_l$ | 8.520 d | 0.090 c | 0.052 c | 0.047 b | 0.188 c | 262.45 a | 130.95 d |
| $L_l T_l W_h$ | 14.135 b | 0.107 b | 0.068 b | 0.048 b | 0.224 b | 198.66 c | 126.60 d |
| $L_l T_h W_l$ | 9.111 c | 0.085 d | 0.044 d | 0.036 d | 0.165 d | 139.87 f | 79.98 f |
| $L_l T_h W_h$ | 19.202 a | 0.114 a | 0.079 a | 0.067 a | 0.259 a | 157.62 e | 94.99 e |

**Notes.**

Abbreviations: HY, hypocotyl; OE, outer epidermal wall; IE, inner epidermal wall; CO, cortical wall; TCW, total cell wall; EP, epidermis; L, light intensity; T, temperature; W, water potential; h, high; l, low.

## DISCUSSION

### Hypocotyl elongation responding to the combined effects of multiple factors

Light, temperature and water potential are often correlated under natural growth conditions and regulate growth and development throughout the life of plants. Previous studies showed that the regulation of light, temperature, and water potential on hypocotyl elongation was interdependent. For example, the classical light response was temperature dependent: the reverse in the response from the inhibition to the promotion of hypocotyl growth by light was induced by a shift in temperature (*Johansson et al., 2014*). In addition, the promotion on hypocotyl elongation by high temperature was light quality dependent and became stronger as F/FR decreased (*Kurepin et al., 2010*). In the present study, we investigated the coordinated regulation of light intensity, temperature, and water potential on hypocotyl elongation (Fig. 1). Low light, high temperature, and high water potential promoted hypocotyl elongation both in isolation and in coordination, which was similar to the previous studies (*Johansson et al., 2014*; *Kurepin et al., 2010*; *Wu et al., 2005*). Under the combined effects of light intensity, temperature, and water potential, the elongation rate incrementally decreased. The rate was relatively quick at 2 d, when the hypocotyl reached approximately 50% of its final length (Fig. 1). The elongation rate decreased by 80% when the 90% final hypocotyl length was reached at 5 d. At 8 d, the final hypocotyl length was reached, and elongation rate decreased. The gradual decrease in the rate contributed to the elongation kinetics of the hypocotyl.

### Contributions of cell elongation and division to hypocotyl growth

Hypocotyl elongation resulted from cell elongation, cell division, or both, depending on species and growth conditions (*Boron & Vissenberg, 2014*; *Kutschera & Niklas, 2013*). Previous studies revealed that hypocotyl elongation in *A. thaliana* occurred primarily by cell elongation with almost no contribution from division (*Gendreau et al., 1997*). Cell division

played a critical role in the hypocotyl elongation of lettuce, radish, and soybean at 0-2 d after germination, and cell elongation played a major role afterwards (*Galli, 1988*). Cell elongation and division were both observed during the hypocotyl elongation of *H. annuus* (*Kutschera & Niklas, 2013*). A previous study showed that cucumber hypocotyl elongated in the absence of cell division when grown in dim light, and cell division appeared under high light (*Lopez-Juez et al., 1995*). In the present study, the contribution of cell elongation and division to hypocotyl growth in *B. rapa* was highly dependent on cell types and growth conditions, and cell division was observed only in the epidermis under low light.

Further study indicated that cell division was only active at the stage of embryogenesis, and cell division was confined to the meristem of the root and stem and the stomatal development region after seed germination in the epidermis of *A. thaliana* and *B. napus* (*Barroco et al., 2005*; *Raz & Koornneef, 2001*). As the receptor of multiple environmental factors, phytochrome B (phyB) played a critical role both in regulating cell elongation in hypocotyl growth and cell division during stomatal development that accounted for the increase in cell layers in the epidermis of *B. rapa* hypocotyl (Fig. 2) (*Casson et al., 2009*; *Wang & Shang, 2019*). Moreover, low light, high temperature, and high water potential significantly promoted cell elongation in hypocotyl by increasing auxin content, which is primarily biosynthesized by *YUCCA8/9* in dicotyledonous plants (*Sun et al., 2012*; *Zhao, 2010*). The expression of *YUCCA8/9* was induced by phytochrome interacting factor 4 (PIF4), which can be degraded after phosphorylation when interacting with phyB in an activated state (*Delker et al., 2014*; *Franklin et al., 2011*). As the emerging hub of environmental signaling pathways in regulating hypocotyl elongation, the protein abundance and activity state of phyB are regulated by light intensity, temperature, and water potential. *Casal & Questa (2018)*, *Legris et al. (2017)* and *Wang & Shang (2019)*. In addition, phyB also regulated hypocotyl elongation by regulating the signaling of hormones, such as IAA and BR. PhyB could interact with auxin/indoleacetic acid (Aux/IAA) proteins to inhibit the signal transduction of IAA (*Xu et al., 2017*). PhyB also attenuated the positive influence of brassinolide (BR) on regulating hypocotyl elongation by repressing the accumulation of PIF4, which interacted with downstream responsive factors of BR, such as BZR1 and BES1 (*Bai et al., 2012*).

## Cell wall: the co-target of light, temperature, and water potential in regulating hypocotyl elongation

Plant cells are surrounded by extensible walls, which allows for the turgor-mediated expansion in cell volume. Newly synthesized polymers should be incorporated into the expanding walls; otherwise, the wall would become thinner until it was broken by turgor pressure (*Derbyshire et al., 2007*; *Refregier et al., 2004*). A previous study showed that changes in wall thickness were relatively complex and highly dependent on cell types, growth conditions, and developmental stages. Different phases of wall thickening, maintenance, and thinning were observed (*Derbyshire et al., 2007*; *Refregier et al., 2004*; *Wolf & Greiner, 2012*). In the present study, the thickness of the OE increased incrementally during hypocotyl elongation. However, the IE and CO displayed phases of becoming thicker, maintaining the thickness, or getting thinner, implying an imbalance of wall

deposition and hypocotyl elongation. The results above were consistent with previous reports (*Derbyshire et al., 2007*; *Fujino & Itoh, 1998*; *Refregier et al., 2004*). Cell wall analysis has been performed in different species, including maize coleoptiles, peas internodes, and *A. thaliana* hypocotyls (*Derbyshire et al., 2007*; *Fujino & Itoh, 1998*). However, this is the first report on the dynamic changes of wall thickness in different cell types as the hypocotyl elongated in *B. rapa*.

The biosynthesis and deposition of the cell wall are regulated by multiple factors. For example, light intensity plays a key role in regulating plant growth by adjusting the deposition of cellulose, hemicellulose, and lignin (*Le Gall et al., 2015*). Constant or transitory high temperatures can induce a series of physiological and biochemical changes in plant growth by inducing an increase in hemicellulose deposition and a decrease in pectin accumulation (*Lima et al., 2013*; *Suwa et al., 2010*). Under water-deficit and salt stresses, cellulose and hemicellulose contents remained unchanged, but pectin content markedly increased (*An et al., 2014*; *Muszyńska, Jarocka & Kurczynska, 2014*). In the present study, the contribution of the wall components to the dynamic changes in wall thickness was highly dependent on the light intensity, temperature, and water potential. Light intensity had significant effects on cellulose, hemicellulose, and pectin deposition, and high temperature and high water potential had significant effects on pectin content. Pectin is the most abundant component of the primary wall in eudicot plants and regulates hypocotyl growth in response to low light, high temperature, and high water potential by influencing wall extensity (*Wolf & Greiner, 2012*; *Xiao, Somerville & Anderson, 2014*; *Xiao et al., 2017*). The gradients in pectin contributed to the gradient elongation rate of the hypocotyl, with higher rates in the apex and lower rates in cells near the base (*Goldberg, Morvan & Roland, 1986*; *Phyo et al., 2017*). The results above indicated that the cell wall may be the target of multiple factors in regulating hypocotyl elongation. Light intensity, temperature, and water potential regulated hypocotyl elongation by affecting wall deposition, especially the deposition of pectin.

## Imbalance of wall deposition and hypocotyl elongation contributed to dynamic changes in wall thickness

As the hypocotyl elongated, the volume of the hypocotyl and the cell wall both increased. However, the increase in the volume of the cell wall did not keep pace with that of hypocotyl, so the increase in cell size was achieved by stretching the existing wall, and then wall thickness decreased (Table 4). At the final length, the hypocotyl was 1.93-fold longer under low light than under high light, but the thickness of the OE, IE, and CO was reduced by 0.19-, 0.35-, and 0.35-fold, respectively. The increase in hypocotyl length induced by high temperature was 1.29-fold, while the decrease in the thickness of the OE, IE, and CO was 0.16-, 0.09-, and 0.14-fold, respectively. In addition, high water potential promoted hypocotyl elongation by 1.18-fold, and the OE, IE, and CO decreased by 0.14-, 0.18-, and 0.14-fold, respectively. The results above indicated that the imbalance of hypocotyl elongation and wall deposition contributed to the changes in wall thickness. The changes in wall thickness could subsequently influenced hypocotyl elongation, and the elongation rates declined as wall thickness increased.

In the present study, we analyzed the combined effects of light intensity, temperature, and water potential on plant growth using hypocotyl as a model system. The interaction of these factors was particularly evident at the early stage of seedling growth, which indicated that more attention should be paid to temperature and water control in the early stage of seedling growth to prevent them from being leggy in the process of vegetable production and planting. In addition, this description of the dynamic changes in wall thickness provided a sound baseline for the identification of key stages of hypocotyl elongation and wall biosynthesis.

## CONCLUSIONS

Light intensity, temperature and water potential regulated the hypocotyl growth of *B. rapa* both in isolation and combination. Low light, high temperature, and high water potential promoted hypocotyl growth by regulating cell elongation in a process that was tightly controlled by the cell wall. The thickness of OE, IE, and CO changed variously, namely, thickening-thinning-thickening, thickening-thinning-maintaining, and thickening-maintaining-thinning, respectively. Further analysis revealed that the imbalance in wall deposition and hypocotyl elongation contributed to dynamic changes in wall thickness. Light intensity, temperature and water potential modulated cell wall deposition by regulating pectin biosynthesis. In conclusion, light intensity, temperature, and water potential regulated hypocotyl elongation of *B. rapa* by influencing wall deposition, especially the deposition of pectin.

### Funding
This work was supported by grants from the National Natural Science Foundation of China (31172001), the China Agriculture Research System (CARS-25), the Special Fund for Agro-Scientific Research in the Public Interest of China (201303014), and the Science and Technological Innovation Program of Chinese Academy of Agricultural Sciences (CAAS-ASTIP-IVFCAAS). The funders had no role in study design, data collection and analysis, decision to publish, or preparation of the manuscript.

### Grant Disclosures
The following grant information was disclosed by the authors:
National Natural Science Foundation of China: 31172001.
China Agriculture Research System: CARS-25.
Special Fund for Agro-Scientific Research in the Public Interest of China: 201303014.
Science and Technological Innovation Program of Chinese Academy of Agricultural Sciences: CAAS-ASTIP-IVFCAAS.

### Competing Interests
The authors declare there are no competing interests.

## Author Contributions

- Hongfei Wang conceived and designed the experiments, performed the experiments, analyzed the data, prepared figures and/or tables, authored or reviewed drafts of the paper, and approved the final draft.
- Qingmao Shang conceived and designed the experiments, analyzed the data, authored or reviewed drafts of the paper, and approved the final draft.

## Data Availability

The raw data is available in the Supplemental Files.

## Supplemental Information

Supplemental information for this article can be found online at http://dx.doi.org/10.7717/peerj.9106#supplemental-information.

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
