# Peer review of "The combined effects of light intensity, temperature, and water potential on wall deposition in regulating hypocotyl elongation of Brassica rapa"

_PeerJ, doi:10.7717/peerj.9106_

## Round 0.1 · original submission · Major Revisions

When submitting the revised version of your manuscript, please provide detailed responses to reviewers' remarks. Mainly, I would like to draw your attention on comments made by reviewer #2 about the statistical analyses of data about the effect of water potential/temperature/light intensity on the genes' expression.

Reviewer 1 ·

Basic reporting

1. When binary comparisons described, the order of comparison is not clear. If all binary comparison were done in the same order as in lines 188, 192, 195 this should be noted in the text. In addition, the statistical test which done base on these comparisons is not clear. e.g. Line 228: “The three environmental factors significantly influenced the elongation of epidermal …”, what statistical test used and what are the parameters?
2. In the results section, some of the measurements are described twice, once in the figure/table and a second time is in the main text. My general suggestion is, if the measurements are clearly described in the figure there is no need to specify it again in the text. These will make the reading more fluent. e.g. lines 177-184.
3. When OE, IE and CO used in the results section for the first time (line 247) the full name needs to be added.
4. In Fig. 3 adding letters will help to clarify it. My suggestion will be to add A for the OE panel and B for the CO and IE panel. Describe the different letters in the figure legend.
5. In table 4, need to describe what is TCW (column 6).
6. In the paragraph starting at line 318, it’s not clear which comparison was done in order to get the fold change. For example: Is the increase of 1.92 in response to low light is the average of all the low light conditions (LlThWh, LlThWl, LlTlWl,LlTlWh) relative to all high light conditions (LhThWh,LhThWl,LhThWl,LlTlWh)? This need to be clarified.
7. I couldn’t find the legends for all the Sup. Fig. They need to be added.
8. In Fig S4 there is no title for the y-axis.

Experimental design

Most of the cell wall measurements come from the middle of the hypocotyl. This could raise some concern that it does not reflect the whole hypocotyl. Since the author clearly described it, and explain the rationale of doing it there, it can stay as it is.

Validity of the findings

1. I have a concern regarding the author conclusion “ Light intensity had significant effects on cellulose, hemicellulose, and pectin deposition, but high temperature and high water potential only had significant effects on pectin content”. This conclusion was based on results from Table S2. However, the analysis done in this table is unclear, showing the formula used for the Anova test will be helpful. In addition, temperature and water potential had shown some effect on cellulose and hemicellulose in some tested conditions: e.g. cellulose and hemicellulose were affected by water potential when LlTlWl compared with LlTlWh. and by temperature when LlTlWl compared with L1ThWl. Therefore this conclusion should be phrased differently or explained in a more convincing way.

Additional comments

This manuscript by Wang and Shang showed a detailed phenotypic analysis of hypocotyl elongation, cell wall thickness and composition, under combined effects of light intensity, temperature and water potential. This approach reveals that these three environmental factors are effecting hypocotyl cell elongation independently and in combination. The authors suggest that low light represses wall deposition by influencing the accumulation of cellulose, hemicellulose and pectin while, high temperature and high water potential effects pectin accumulation only.
The authors collect an extensive amount of high quality and valuable data, the question addressed is clearly outlined and the manuscript is generally written in a logical order.
There are a few points in which clarity are needed as described above.

Reviewer 2 ·

Basic reporting

The outline of the article is clear and easy to follow. However, the text has spelling and grammatical mistakes even though an editorial certificate has been provided.
Sufficient context is provided to understand the article.
The figures and tables and their legends are self-explanatory.

Experimental design

1. Line 103 - The authors study the middle segment of the hypocotyl based upon previous studies that showed that the cells at the mid- and apical-segment contributed to hypocotyl elongation. It would help to know if the environmental conditions in this study affect hypocotyl elongation in a similar way before assuming that the mid-segment is the only part of the hypocotyl affected.
2. Line 122-136 - It is very difficult and confusing to understand the measurements here. It would help to define the transverse wall, and the outer and the inner epidermal cell wall here.
3. Furthermore, a supplementary figure showing the parameters of the cell wall (i.e. length, breadth, perimeter, etc.) taken into account for the calculations of the volume.
4. What is the meaning of "the number of IE/OE"? Are they counting the number of walls?

Validity of the findings

1. Are the effects of the environmental factors significant under high and low conditions? It would help to provide the p-values in all such comparisons. For example, how significant is the effect of temperature on the hypocotyl elongation, irrespective of the light intensity and the water potential? And likewise for the other comparisons. From the text, it seems like the authors have concluded the significance by eyeballing.
2. Line 303-314 - The expression of the genes included in the text do not follow the trend in Figure S8. For example, the expression of the genes CSCL and MUR3 are not inhibited and GAUT7 was not induced at high temperatures. The same goes for the rest of the genes.
3. Where are the statistics showing the effect of water potential/temperature/light intensity on the genes' expression?
4. Line 375 - Gene name - YUCCA or YUC

Reviewer 3 ·

Basic reporting

Figure 1, 2, and 3: The legends should include what the L, T, W, and subscript h, i are representing.

Figure 4: For clarity makes the bar color white and black for the lowest and highest hypocotyl elongation, respectively. Then follow a gradient of pattern for all other bars. That will help to co-relate things immediately.

Experimental design

The knowledge gap and the research question are not clearly mentioned in the abstract. Moreover, the abstract is loaded with redundant information that is causing confusion to understand what the main observation and conclusion are.

Validity of the findings

No Comments

Additional comments

Thank you for the nicely done and well-written manuscripts.

Reviewer 4 ·

Basic reporting

The present manuscript is the revised version of the previously submitted manuscript. The authors have substantially improved the language of their reporting and the content is easy to follow and understand. They have provided sufficient references and background for their work. The authors have provided the shorthand of their treatments for example HhLlWh which is easy to understand. However, they have not added these acronyms to when they are explaining their data leaving the readers to guess which data (for example see lines 222-227, page 6) corresponds to what treatment. So, it would be appropriate if the authors state somewhere in the manuscript as to the order of the treatment that they are going to follow when they are reporting their data or provide the acronyms as they describe their data.

Experimental design

The authors have provided the statistical analysis for most of their data reported which is important to show the significance of the data.
The authors have now provided in detail the formula that they have used to measure the cell wall volume which is helpful in knowing the approach they have taken in the measurments. The formula suggests that the authors are reporting the cell wall volume for the entire hypocotyl length as they use the number of cells in the hypocotyl in their analysis for example: the formula of calculating cortical cells (CO) is perimeter of the CO by its number, by the thickness of the CO, and by the hypocotyl length.. It would be helpful if the authors provide the number of cells for cortex and inner epidermal wall columns in Table 4.

Validity of the findings

no comment

---

## Round 0.2 · Major Revisions

According to one reviewer, the data analysis approach is not yet sufficient to support the working hypothesis. I suggest considering a statistical model that allows you to evaluate both the effects of individual factors, both their interactions and the random effect.

Reviewer 1 ·

Basic reporting

In the raw data file, the tab “cell wall extension” has some words which are not in English.

Experimental design

no comment

Validity of the findings

Making the changes below will make it easier to understand:

- Line 302-304, clarify this part by splitting it into 2 sentences, and refer to Table S2 after: “The influence of temperature on pectin deposition reached significant level…”. Also refer to Table S2 after line 297 “Low light significantly repressed cellulose, hemicellulose, and pectin deposition…”
- Line 307-309: When described the effect of water potential “…and the effect on pectin accumulation did not reach significant level under the background of high light × low water potential (Table 3; P < 0.05; Duncan’s multiple range test).” Since it does reach a significant level overall (Table S2), as also mentioned in the abstract, adding a sentence that claims it, like done for the influence of temperature is necessary.

Additional comments

The revised version of the manuscript “The combined effects of light intensity, temperature, and water potential on wall deposition in regulating hypocotyl elongation of Brassica rapa”, addressed all the points from my first review. They now described their findings in a better way especially in explaining the statistical analysis they did.

Reviewer 2 ·

Basic reporting

The language needs to be corrected. It can be difficult for a non-native speaker but it would get it corrected professionally.

Experimental design

Thank you for making a figure to explain the volume measurements. It makes it easy to understand the article and its finding better.

Validity of the findings

The statistical question has not been answered yet.
The statistical analyses do not prove that the effect of one stimulus. They represent the significance of the combinatorial effect. Without proper statistical tests, the conclusions that the stimuli regulate cell wall thickness in isolation does not hold. Probably going through some R-based programs to do the statistical tests would help. But this is important.

---

## Round 0.3 · Minor Revisions

Just a few minor changes are still needed: the supplementary figures and tables do not have legends to explain the content.

Reviewer 2 ·

Basic reporting

Clear English used.
Enough literature provided to help the paper.
The supplementary figures and tables do not have legends to explain the content.

Experimental design

No comments.

Validity of the findings

I thank the authors for taking care of the statistical analyses. It helps strongly support the results of the experiments included.

---

## Round 0.4 · accepted · Accept

The indications of the reviewers were correctly taken into account by the authors. I think the article is now suitable for publication.